https://doi.org/10.1038/s41467-020-17217-1　　**OPEN**

# Gene regulatory network inference from sparsely sampled noisy data

Atte Aalto[1], Lauri Viitasaari[2], Pauliina Ilmonen[3], Laurent Mombaerts [1] & Jorge Gonçalves [1,4 ✉]

The complexity of biological systems is encoded in gene regulatory networks. Unravelling this intricate web is a fundamental step in understanding the mechanisms of life and eventually developing efficient therapies to treat and cure diseases. The major obstacle in inferring gene regulatory networks is the lack of data. While time series data are nowadays widely available, they are typically noisy, with low sampling frequency and overall small number of samples. This paper develops a method called BINGO to specifically deal with these issues. Benchmarked with both real and simulated time-series data covering many different gene regulatory networks, BINGO clearly and consistently outperforms state-of-the-art methods. The novelty of BINGO lies in a nonparametric approach featuring statistical sampling of continuous gene expression profiles. BINGO's superior performance and ease of use, even by non-specialists, make gene regulatory network inference available to any researcher, helping to decipher the complex mechanisms of life.

[1] Luxembourg Centre for Systems Biomedicine, University of Luxembourg; 6 avenue du Swing, 4367 Belvaux, Luxembourg. [2] Department of Mathematics and Statistics, University of Helsinki; P.O. Box 68, Gustaf Hällströmin katu 2b, 00014 Helsinki, Finland. [3] Department of Mathematics and Systems Analysis, Aalto University School of Science; P.O. Box 11100, 00076 Aalto, Finland. [4] Department of Plant Sciences, University of Cambridge; Downing Street, Cambridge, UK. ✉email: jorge.goncalves@uni.lu

Cellular functions are controlled by the mechanism of gene expression, and the regulatory interconnection of genes. Knowledge of the regulatory structure enables understanding of biological mechanisms, and can, for example, lead to discoveries of new potential targets for drugs. Interactions between genes are typically represented as a gene regulatory network (GRN) whose nodes correspond to different genes, and a directed edge denotes a direct causal effect of some gene on another gene. The usual research aim is to infer the network topology from given gene expression data. Classically, GRN inference has been based on analysing steady-state data corresponding to gene knockout experiments, where one gene is silenced and changes in the steady-state expressions of other genes are observed. However, carrying out knockout experiments on a high number of genes is costly and technically infeasible. Moreover, it can be difficult to know if the system really is in a steady state when it is measured. By contrast, methods that infer GRNs from time series data can infer the networks on a more global level using data from few experiments. Therefore, GRN inference from time series data has gained more and more attention recently[1–5].

Modelling dynamical systems from time series data are a problem with long history, in particular in the fields of mechanical, electrical, and control engineering. Inference of GRNs, however, adds further modelling challenges since data collection is expensive and technically demanding. Typically gene expression data have low sampling rates and relatively small amount of data. Moreover, GRNs have a high number of genes with complex, nonlinear regulatory mechanisms.

Several GRN inference problems, from different types of data, have been posed as competitive challenges by the "Dialogue for Reverse Engineering Assessments and Methods" (DREAM) project[2,6]. In addition, different methods have been compared in review articles[1,2]. Many different approaches have been taken to solve the problem, such as graphical models[7], Bayesian network models[8], information theory methods[9,10], neural models[5], and so on. The main focus of this article is on methods that are based on fitting an ordinary differential equation (ODE) model to the observed gene expression time series data.

Most ODE-based methods transform the system identification problem into an input–output regression problem where the inputs are the measured gene expression values and outputs are their derivatives that are estimated from the data. Derivative estimation can be based on simple difference approximation[3,4,11,12], spline fitting[13], Gaussian process fitting[14,15], or regularised differentiation[16,17]. The regression problem is then solved by linear methods[13], fitting

mechanistic functions[12,15], or a user-defined library of nonlinear functions[4,16,18,19]. Also nonparametric machine learning techniques have been used, such as random forest[3] and Gaussian process regression[11,14,20]. A method based on Gaussian process regression, called "Causal structure identification" (CSI)[14,20] was the best performer in a comparison study for network inference from time series data[1].

Another ODE-based approach is to introduce a simple enough model class, from which trajectories can be directly simulated and compared to the measured data. Such strategy does not suffer badly from the low sampling rate, but the model class cannot be too complex, and it might be too restrictive to capture the behaviour of the real system. Linear dynamics have been proposed together with a Kalman filter state estimation scheme[21]. The winner of the DREAM4 in silico network inference challenge was a method called "Petri nets with fuzzy logic" (PNFL)[22], whose model class consists of fuzzy membership functions.

This article presents BINGO (Bayesian Inference of Networks using Gaussian prOcess dynamical models). The novelty of BINGO is the introduction of statistical trajectory sampling. This enables the use of a flexible nonparametric approach for modelling the nonlinear ODE on continuous gene expression trajectories. BINGO is based on modelling gene expression with a nonlinear stochastic differential equation where the dynamics function (or drift function), is modelled as a Gaussian process. This defines gene expression as a stochastic process, whose realisations can be sampled using Markov chain Monte Carlo (MCMC) techniques. The key to overcome low sampling frequency is to sample the trajectory also between measurement times. The method pipeline is illustrated in Fig. 1. The Gaussian process framework is very flexible, which enables capturing complex nonlinear regulations between genes. Overfitting is avoided by integrating out all hyperparameters in the model, and concentrating solely on the network topologies, that are given a sparsity promoting prior.

While BINGO is not the first GRN inference method to utilise Gaussian processes to model the dynamics function in an ODE model, the existing approaches[11,14] treat the problem as an input–output regression problem, with gene expression derivatives estimated directly from the time series data. This approach yields a probability distribution for the derivatives, whereas we obtain a probability distribution for the continuous gene expression trajectories. This allows by-passing the derivative estimation, which can be a serious source of errors when the sampling rate is low.

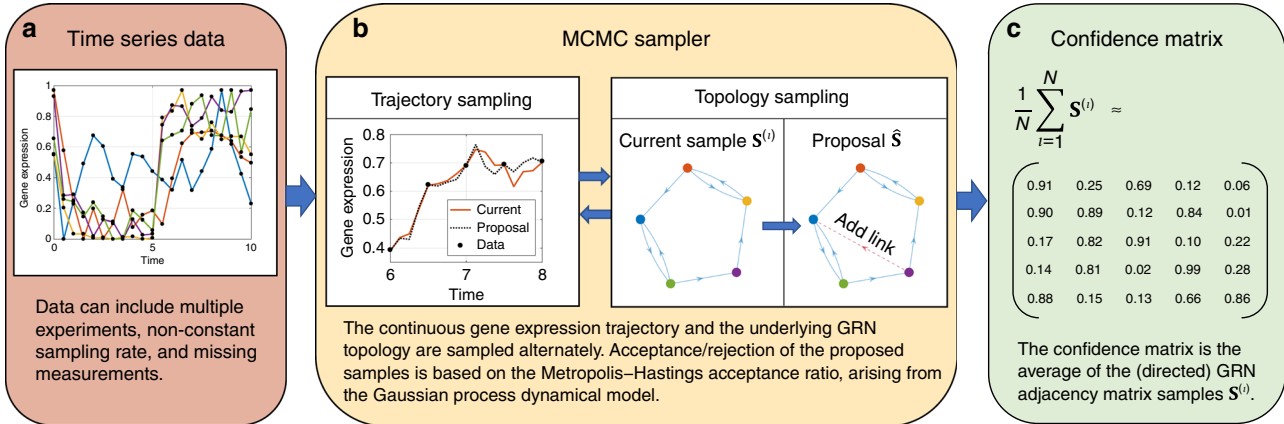

**Fig. 1 The pipeline of BINGO. a** BINGO can utilise time series data originating from any experimental technique. **b** A proposal trajectory sample $\hat{\mathbf{x}}$ is drawn by perturbing slightly the current sample $\mathbf{x}^{(l)}$. The proposal is then accepted ($\mathbf{x}^{(l+1)} = \hat{\mathbf{x}}$) or rejected ($\mathbf{x}^{(l+1)} = \mathbf{x}^{(l)}$) based on the Metropolis–Hastings acceptance ratio. The GRN topology proposal $\hat{\mathbf{S}}$ is constructed by adding or removing one link to/from the current topology $\mathbf{S}^{(l)}$. **c** Output of BINGO is a confidence matrix of posterior probabilities for the existence of links.

BINGO is validated by comparing it with state-of-the-art network inference methods using data from the DREAM4 in silico network inference challenge[2,23,24]. Moreover, the effects of sampling frequency and process noise are studied by applying the method to simulated data of the circadian clock of *Arabidopsis thaliana*[25]. Finally, BINGO's performance on real experimental data is demonstrated by applying it to a dataset from the synthetic network IRMA[26], and by using it on drug target identification. BINGO outperforms state-of-the-art network inference methods in inference from time series data.

## Results

**Gene expression modelled with differential equations.** Gene expression time series data $\mathbf{Y} = \{\mathbf{y}_0, \dots, \mathbf{y}_m\}$ are modelled as samples from a continuous trajectory,

$$\mathbf{y}_j = \mathbf{x}(t_j) + \mathbf{v}_j, \tag{1}$$

where $\mathbf{v}_j$ represents measurement noise. The continuous trajectory is assumed to satisfy a nonlinear stochastic differential equation

$$d\mathbf{x} = \mathbf{f}(\mathbf{x})dt + d\mathbf{w}, \tag{2}$$

where $\mathbf{w}$ is some driving process noise. Here, $\mathbf{x}$ is an $\mathbb{R}^n$-valued function, and thus also $\mathbf{f}$ is vector-valued,

$$\mathbf{f}(\mathbf{x}) = \begin{bmatrix} f_1(x_1, \dots, x_n) \\ \vdots \\ f_n(x_1, \dots, x_n) \end{bmatrix}. \tag{3}$$

If function $f_i$ depends on $x_j$, in the corresponding GRN there is a link from gene $j$ to gene $i$. The task in GRN inference is to discover this regulatory interconnection structure between variables.

BINGO is based on a continuous-time version of the so-called Gaussian process dynamical model (GPDM). Essentially, the dynamics function $\mathbf{f}$ in Eq. (2) is modelled as a Gaussian process with some covariance function[27]. This defines $\mathbf{x}$ as a stochastic process with probability distribution $p(\mathbf{x}|\theta)$, where $\theta$ denotes hyperparameters of the Gaussian process—including the underlying GRN topology. Trajectory realisations from the conditional probability distribution

$$p(\mathbf{x}|\theta, \mathbf{Y}) \propto p(\mathbf{Y}|\mathbf{x}, \theta)p(\mathbf{x}|\theta), \tag{4}$$

can be sampled using MCMC techniques. Here $p(\mathbf{Y}|\mathbf{x}, \theta)$ is the measurement model arising from Eq. (1). This trajectory sampling can be regarded as statistical interpolation where the trajectories are required to be consistent with both the measured data and the underlying dynamical system. No additional information is imputed in generating trajectory samples. The decoupling of the trajectory $\mathbf{x}$ and the measurements $\mathbf{Y}$ in Eq. (4) enable easy treatment of irregularities in time series data, such as missing measurements and non-constant sampling frequency.

**GPDM for network inference.** The discrete-time GPDM[28] (a.k.a. Gaussian process state space model[29,30]) is based on GP latent variable models[31]. It is an effective tool for analysing time series data produced by an unknown dynamical system, or in the case the system is somehow too complicated to be presented using classical modelling techniques. In the original paper[28], the method was used for human motion tracking from video data. Motion tracking problems remain the primary use of GPDMs[32,33], but other types of applications have emerged as well, such as speech analysis[34], traffic flow prediction[35], and electric load prediction[36]. BINGO is based on continuous-time GPDM, which is introduced

in this article, and some theory of GPDMs is presented in Supplementary Notes 1–3.

A Gaussian process $f$ is a stochastic process, that is, a function whose values at any given set of input points form a random variable. Gaussian processes can be seen as a generalisation of normally distributed random variables. A Gaussian process is defined on some index set $\Xi$ so that for any finite collection $\overline{\xi} = [\xi_1, \dots, \xi_N]^\top \in \Xi^N$, the vector $[f(\xi_1), \dots, f(\xi_N)]^\top$ is normally distributed. Just like in the finite-dimensional case, properties of a Gaussian process are completely comprised in a mean function $m(\xi) = \mathbb{E}(f(\xi))$, and a covariance function $k(\xi_1, \xi_2) = \mathrm{Cov}(f(\xi_1), f(\xi_2))$. Then it holds that

$$[f(\xi_1), \dots, f(\xi_N)]^\top \sim \mathcal{N}\left(\mathbf{m}(\overline{\xi}), \mathbf{K}(\overline{\xi}, \overline{\xi})\right), \tag{5}$$

where $\mathbf{m}(\overline{\xi}) = [m(\xi_1), \dots, m(\xi_N)]^\top$ and $\mathbf{K}(\overline{\xi}, \overline{\xi})$ is an $N \times N$ matrix whose element $(i, j)$ is $k(\xi_i, \xi_j)$.

Given noisy samples $\eta_j = f(\xi_j) + v_j$, for $j = 1, \dots, N$ with $v_j \sim \mathcal{N}(0, r)$, the value $f(\xi)$ at a generic point $\xi$ can be approximated with the conditional expectation, which can be expressed analytically

$$\mathbb{E}(f(\xi)|\overline{\eta}) = m(\xi) + \mathbf{K}(\xi, \overline{\xi})\left(\mathbf{K}(\overline{\xi}, \overline{\xi}) + r\mathbf{I}\right)^{-1}(\overline{\eta} - \mathbf{m}(\overline{\xi})). \tag{6}$$

This is the basis of Gaussian process regression which is extensively used in nonlinear regression problems in the field of machine learning[27]. Its popularity is based on its analytical tractability and solid roots in probability theory. For example, it is possible to obtain confidence intervals for predictions of GP regression. In GPDM, thanks to the analytical tractability of the GP framework, it is possible to obtain the probability distribution $p(\mathbf{x}|\theta)$ for solutions of the stochastic differential Eq. (2), which is a vital component of BINGO.

The GP framework can handle combinatorial effects, meaning nonlinearities that cannot be decomposed into $f(x_1, x_2) = f_1(x_1) + f_2(x_2)$. This is an important property for modelling a chemical system—such as gene expression—where reactions can happen due to combined effects of reactant species. For example, the dynamics corresponding to a chemical reaction $x_1 + x_2 \rightarrow x_3$ cannot be modelled by $\dot{x}_3 = f_1(x_1) + f_2(x_2)$.

In a classical variable selection problem from input–output data $\eta_j = f(\xi_j) + v_j$, for $j = 1, \dots, N$, where $\xi_j \in \mathbb{R}^n$, the task is to find out which components of $\xi$ does the function $f$ depend on. In the GP framework, variable selection can be done by a technique known as "automatic relevance determination"[37,38], based on estimating the hyperparameters of the covariance function of $f$. For BINGO, we have chosen the squared exponential covariance function for each component $f_i$:

$$k_i(\mathbf{x}, \mathbf{z}) = \gamma_i \exp\left(-\sum_{j=1}^n \beta_{i,j}(x_j - z_j)^2\right). \tag{7}$$

The hyperparameters $\beta_{i,j} \geq 0$ are known as inverse length scales, since $1/\sqrt{\beta_{i,j}}$ corresponds to a distance that has to be moved in the direction of the $j^{\text{th}}$ coordinate for the value of function $f_i$ to change considerably. If $\beta_{i,j} = 0$, then $f_i$ is constant in the corresponding direction. The mean function for $f_i$ is $m_i(\mathbf{x}) = b_i - a_i x_i$ where $a_i$ and $b_i$ are regarded as nonnegative hyperparameters corresponding to mRNA degradation and basal transcription, respectively.

Network inference using BINGO is based on Bayesian estimation of parameters $\beta_{i,j}$: $p(\theta|\mathbf{Y}) \propto p(\mathbf{Y}, \theta) = \int p(\mathbf{Y}, \mathbf{x}, \theta)d\mathbf{x} = \int p(\mathbf{Y}|\mathbf{x}, \theta)p(\mathbf{x}|\theta)p(\theta)d\mathbf{x}$. Again $\theta$ consists of all hyperparameters in the method, including the interesting parameters $\beta_{i,j}$. The integral with respect to the continuous trajectory $\mathbf{x}$ is done by sampling trajectories from the distribution $p(\mathbf{Y}|\mathbf{x}, \theta)p(\mathbf{x}|\theta)$, as described

above. Also, all the hyperparameters are sampled, including $\beta_{i,j}$ that are given zero-preferring priors by using an indicator matrix **S**, which is the adjacency matrix of the corresponding directed graph structure. BINGO's output is the average of the indicator samples (see Fig. 1). The elements of this matrix converge to posterior probabilities of the existence of corresponding links in the GRN— of course subjected to the validity of the underlying (implicit) assumptions. BINGO can be applied on time series data of gene expression produced by any experimental technique.

**Benchmarking.** BINGO has been benchmarked using data from the DREAM4 in silico network challenge, simulated data from the circadian clock of the plant *Arabidopsis thaliana* with different sampling rates and process noise levels, and the IRMA in vivo dataset. The DREAM4 and IRMA data represent systems' adaptation to external perturbations, whereas the circadian clock is a naturally dynamic, oscillating system. In all benchmark experiments, BINGO is compared with four recent methods, dynGENIE3[3], iCheMA[15], ARNI[4], and GRNTE[9] (see Supplementary Note 9 on the implementation of ARNI and dynGENIE3). They are designed for inference from time series data. In addition, DREAM4 and IRMA datasets have been used in other benchmarking studies, and some results found in the literature and the best performers in the DREAM4 challenge have been included in the comparison. Standard classifier scores are used for the comparison, namely the area under the receiver operating characteristic curve (AUROC) and the area under the precision-recall curve (AUPR). The AUROC score gives equal weight to all predictions, whereas the AUPR is more sensitive to the correctness of high confidence predictions. GRNs are typically sparse, causing a class imbalance between true and false links. The low confidence predictions therefore contain a high number of false positives, rendering the predictions with low confidence less important. The AUPR should therefore be considered as the primary performance metric. Self-regulation is always excluded in the computation of the AUROC and AUPR scores as in the DREAM4 challenge. Aggregated results of the benchmark cases are illustrated in Fig. 2.

*DREAM4 in silico network challenge:* The challenge consists of ten network inference tasks—five tasks with network size 10 and five with size 100[2,23,24,39]. The simulated data consist of five time series for each 10-gene network and ten time series for each 100-gene network. These time series simulate microarray experiments where static perturbations have been applied on some genes for the first half of the recording time. The time series illustrate the system's adaptation to the perturbation, and its relaxation when the perturbation is removed. Each time series consists of 21 time points. In addition, steady-state values are provided as well as

gene knockout and knockdown data corresponding to each gene. The 10-gene challenge provides multifactorial data, corresponding to steady-state values under mild perturbations on the basal transcription rate. This corresponds to data collected from different cells, for example.

BINGO is first compared with the aforementioned methods using the time series data, and then with the challenge best performers using all available data. The use of steady-state data from knockout/-down experiments by BINGO is discussed in Supplementary Note 5. The best performer in a recent comparison of network inference methods from time series data[1] (in terms of average AUPR value) was a method called "Causal Structure Identification"[14,20], which is also based on Gaussian process regression. Its discrete-time version is included in the comparison, since its performance was better. The 10-gene challenge winner was "Petri Nets with Fuzzy Logic"[22]. The 100-gene challenge winner[40] used only the knockout data. A similar method, the "median-corrected Z-score" (MCZ)[41], achieved the second highest score in the 100-gene challenge. Any method inferring networks from time series data can be combined with a method inferring GRNs from steady-state data[41], such as the MCZ. Unfortunately, the MCZ requires knockouts or knockdowns of all genes, which can hardly be expected in a real experiment. Nevertheless, the combinations dynGENIE3*MCZ and BINGO*MCZ are included in the full data comparison. The scores for the combined methods are the products of the individual scores, favouring links that score high in both methods. It should be noted that BINGO (as well as the PNFL) can utilise also partial knockout data together with time series data. The results on DREAM4 data are summarised in Supplementary Table 4.

BINGO consistently outperforms other methods by a large margin (with the exception of network 3) in GRN inference from time series data in the 10-gene case. When using all data from the challenge, BINGO scores slightly higher (average AUPR) than the DREAM4 10-gene challenge winner PNFL. The average scores are very close to each other but in the different networks there are some rather significant differences. BINGO reaches a fairly high AUPR in network 2, which seemed to be very difficult for all challenge participants. The best AUPR for network 2 among the challenge participants was 0.660, and the PNFL's 0.547 was second highest[39]. The poor performance of most methods with network 2 is attributed to low effector gene levels in the wild type measurement[22]. In contrast, BINGO's performance is less satisfactory with network 3, where the PNFL achieves almost perfect reconstruction. This might be due to a fairly high indegree (four) of two nodes in the true network. Only one out of eight of these links gets higher confidence value than 0.5 assigned by BINGO. Based on Supplementary Table 4 and (ref. 1, Table 1),

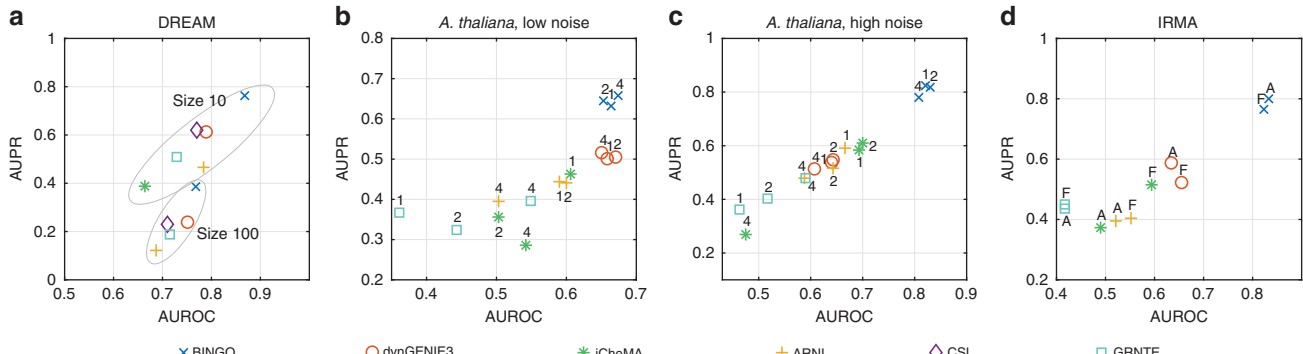

**Fig. 2 Results on benchmark experiments. a** The DREAM results consist of average values over the five networks using the time series data only. **b**, **c** The results on the circadian clock of *Arabidopsis thaliana* consist of averages over ten replicates. The results for different sampling rates (1 h/2 h/4 h) are shown separately. **d** In the IRMA results, both full data (F) and averaged data (A) results are shown separately.

the impact of knockout data is largest in network 3. PNFL perhaps makes better use of these data. Also the BINGO*MCZ combination scores fairly well with network 3, but in network 2 it loses clearly to BINGO applied to all data directly.

As in the 10-gene case, BINGO outperforms its competitors by a clear margin in all five networks when inferring the 100-gene networks from time series data alone, and in fact, it scores slightly higher than the DREAM4 challenge winner. iCheMA is excluded due to its poor scalability to high dimension. When using all data, the combination BINGO*MCZ is the best performer, tied with the combination dynGENIE3*MCZ. It seems that with the 100-gene network, BINGO cannot always combine different types of data in an optimal way. This may be due to the large number of steady-state points where the dynamics function **f** should vanish. This hypothesis is supported by the deterioration of results when also the knockdown data is included with the knockout and time series data. It should be noted that both the DREAM4 winner as well as the MCZ use only the knockout and knockdown data, but they require knockout of every gene, which is hardly realistic in a real experiment.

*Circadian clock of Arabidopsis thaliana:* Realistic data were simulated from the so-called Millar 10 model of the *Arabidopsis thaliana* circadian clock[25], using the Gillespie method[42] to account for the intrinsic molecular noise. This model has been widely used to study the plant circadian clock and as a benchmark to assess the accuracy of different network inference strategies[15]. It simulates gene expression and protein concentration time series with rhythms of about 24 h. The gene regulatory structure consists in a three-loop feedback system of seven genes and their corresponding proteins whose chemical interactions are described using Michaelis–Menten dynamics. The model has been simulated for 600 h in 24-h light/dark cycles to remove transients. Then, the photoperiodic regime was switched to constant light. Ten replicates were simulated and the first 48 h of the constant light phase was recorded and downsampled to correspond to sampling intervals of 1, 2, or 4 h. The time series therefore consist of 49, 25, or 13 time points depending on the sampling interval. Two datasets were simulated with different levels of process noise, which is due to the random nature of gene expression. Process noise propagates through the network and therefore it should be beneficial to network inference algorithms.

Figure 2b–c shows the mean AUROC/AUPR values for the methods computed from the ten replicates (see also Supplementary Table 5). BINGO and dynGENIE3 are hardly affected by the decreasing sampling frequency. With less process noise, the

AUROC values for these two methods are very close to each other in all cases, but BINGO has somewhat better precision throughout the tested sampling frequencies. With higher process noise, the results of BINGO improve clearly with all sampling rates. For dynGENIE3, iCheMA, and ARNI the performance improves slightly with higher process noise, at least with 1 and 2-h sampling rates. Most methods seem to be able to take advantage of the additional excitation due to process noise. The iCheMA and ARNI results with 4-h sampling rates and 2-h sampling rates with low process noise are not much better than random guessing.

*In vivo dataset IRMA:* A synthetic network[26] was constructed with the purpose of creating an in vivo dataset with known ground truth network for benchmarking network inference and modelling approaches. The network is rather small, consisting of only five genes and eight links in the ground truth network (see Supplementary Fig. 1). Nevertheless, this dataset can be used to verify the performance of BINGO using real data.

The IRMA network can be "switched on" and "off" by keeping the cells in galactose or glucose, respectively. The dataset consists of nine transient time series obtained by qPCR technique, where the network is either switched on (five time series) or off (four time series) at the beginning of the experiment. These have been averaged into one switch-on time series (with 16 time points with 20-min sampling interval) and one switch-off time series (with 20 time points with 10-min sampling interval). Typically only the two average time series have been used, but here BINGO is applied both on the two average time series, and on all nine time series.

The resulting AUROC/AUPR scores are presented in Fig. 2d (and Supplementary Table 6). The dataset has been used also in other recent articles presenting methods ELM-GRNNminer[5], and TimeDelay-ARACNE[43]. They only report one network structure as opposed to a list of links with confidence scores. Therefore it is not possible to calculate AUROC/AUPR scores for these methods, but their predictions can be presented as points with the ROC and precision-recall curves obtained for BINGO and dynGENIE3, shown in Fig. 3a, b. With such a small network, the AUROC and AUPR values are very sensitive to small differences in predictions. The best predicted network by BINGO using the averaged data has five out of eight links correct, and one false positive. The best predictions from dynGENIE3 with the same data have either four correct links with one false positive or five correct links with three false positives. However, it is not evident if these best predictions can be concluded from the results. With

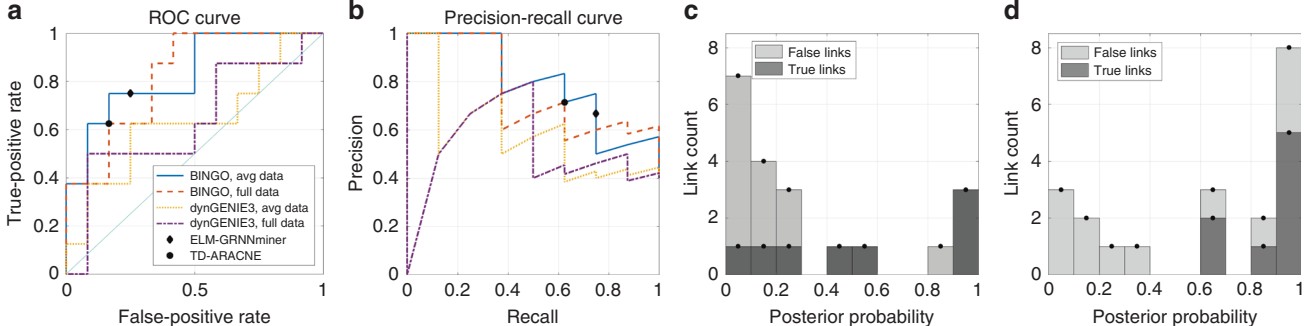

**Fig. 3 Results for in vivo dataset IRMA.** The ROC (**a**) and precision-recall (**b**) curves for BINGO and dynGENIE3 using either all nine time series, or the two averaged time series together with predictions from the ELM-GRNNminer (obtained from ref. [5], Fig. 4) and the TD-ARACNE (from ref. [43], Fig. 5). **c–d** The histograms of posterior probabilities of all links for the averaged (**c**) and full (**d**) IRMA data. In the average data case, the prediction with three true positives and no false positives is obtained with threshold between 0.86 and 0.97. The best prediction with five true and one false positive is obtained with threshold between 0.25 and 0.45. In the full data case, the likely predictions are six true and four false positives with threshold between 0.66 and 0.85, or eight true positives (all) and five false positives with threshold 0.39 and 0.60.

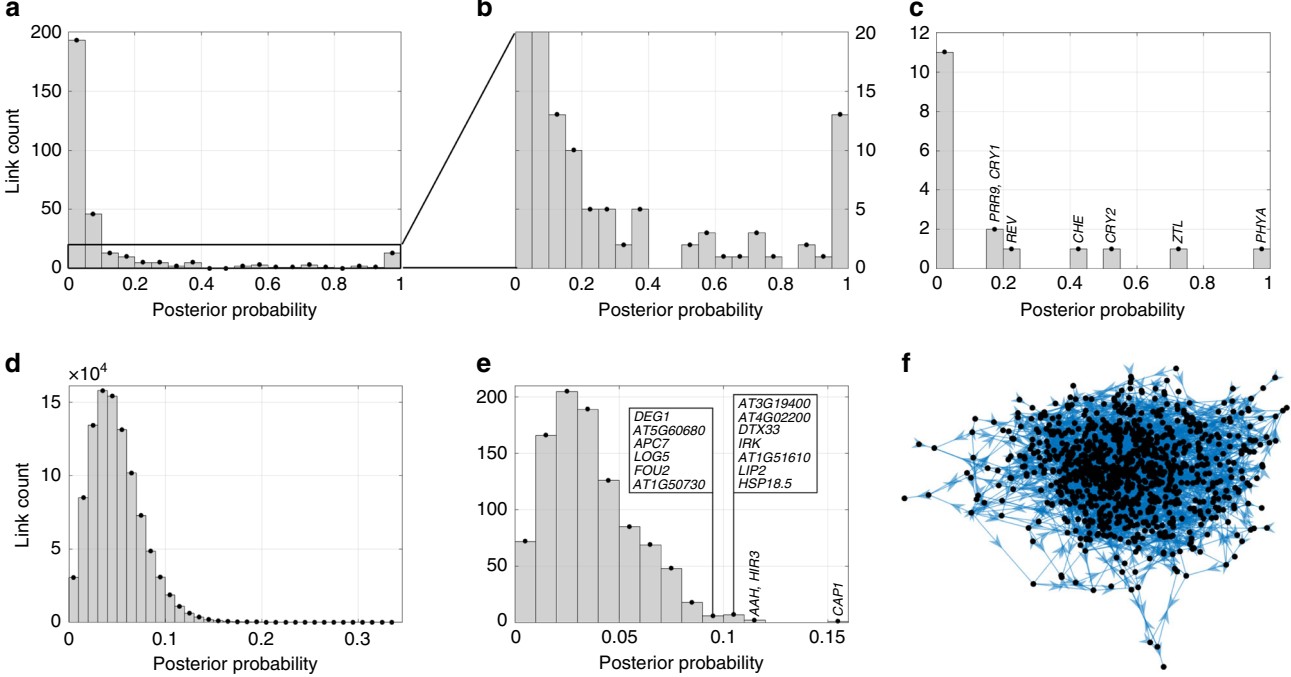

**Fig. 4 Results for drug target identification. a**, **b** Histogram of posterior probabilities of all links for the 17-gene NAM data. **c** Histogram of posterior probabilities for NAM targets for the 17-gene data. **d** Histograms of posterior probabilities of all links for the 994-gene dataset. **e** Histogram of posterior probabilities for NAM targets in the 994-gene dataset. **f** Largest connected component of the network obtained with threshold 0.15 contains 963 genes.

BINGO it is possible to look at the histogram of the posterior probabilities of all possible links, shown in Fig. 3c, d. In the averaged data case, the best prediction with five true links with one false positive stands out relatively well. Using the full data, there are three false positives that get confidence of over 0.9.

**Drug target identification**. To demonstrate BINGO's use in drug target identification, we apply it to a microarray dataset of the circadian clock of *Arabidopsis thaliana*[44]. The data consist of two experiments (each with two replicates): wild type (untreated) and nicotinamide (NAM) treated plants. The study is divided in two parts: a study of 17 known core circadian genes, and a larger study of 994 genes that were periodic in both datasets[44]. For each study, BINGO used all available data at once. To identify direct NAM targets, an external input was added with constant value zero for untreated plants and one for treated plants. Links from NAM to potential targets are modelled exactly as other links in the network. Figure 4a, b shows posterior probabilities for all links between the 17 clock genes. A threshold of 0.85 results in 11 links. A total of eight of those links are included in F14, a well known circadian clock model[45] (note that F14 only has 10 of the 17 clock genes considered here). With a threshold of 0.5 (suggested by the gap in the histogram in Fig. 4b), 11 of the 18 modelled links are also in F14. Figure 4c shows posterior probabilities for NAM targets where seven links have clearly positive posterior probabilities. Of those, *PRR9* and *CRY2* were previously identified as NAM-targets[44].

To illustrate that BINGO can model large scale networks, we applied it to 994 periodic genes in both untreated and NAM plants. The posterior link probabilities are shown in Fig. 4d, and the resulting network is shown in Fig. 4f. This network is likely to reveal new gene interactions, requiring further experimental validations. Figure 4e shows the posterior probabilities histogram for NAM targets. Of those, *CAP1* is a strong candidate that will be experimentally validated in future work.

## Discussion

A nonparametric method BINGO was proposed for GRN inference, which is based on the continuous-time GPDM. The Gaussian process framework has proven very useful in nonlinear regression problems arising in machine learning. Due to the analytical tractability of Gaussian processes, it is possible to obtain a probability distribution for the trajectories of the GPDM. This allows MCMC sampling of the continuous trajectories, thereby bypassing a caveat of estimating derivatives from time series with low sampling frequency—a far too common procedure in existing GRN inference methods.

BINGO was favourably compared to state-of-the-art methods in GRN inference from time series data in various examples. In particular, it was demonstrated that the approach, based on sampling continuous gene expression trajectories, is good for handling time series data with low sampling frequency. Moreover, it was shown that the method can integrate steady-state data with time series data to improve performance. BINGO was also successfully applied on real biological data.

BINGO is computationally heavier than dynGENIE3, for example, which is among the best methods in terms of scalability to large dimensions. However, given the time, effort, and cost of a gene expression experiment, the computation time is hardly as important as the accuracy of predictions, as long as the method is scalable to high enough system dimension. MCMC approach is perfectly parallelisable: independent chains are run on different processors, and the collected samples are pooled together in the end. Parallelisation allows inference of networks of even a couple of thousands of variables. In the NAM target identification, BINGO was applied on a dataset with four time series of 994 genes, in which case the computation time was just under 19 h on a desktop workstation using 20 processors, which could be reduced by resorting to high performance computing.

Recently developed so-called single-cell techniques enable gene expression measurements in one cell resolution for a large number of cells at a time. The cell is destroyed in the measurement process,

and therefore the data consist of ensemble snapshots rather than time series. It is possible to obtain so-called pseudotime series from such data[46,47], and BINGO can be used on such time series —although a small modification in fitting the trajectory samples to the measurements is required, due to the large amount of measurements typically obtained from single-cell measurements. The method can also be integrated with a pseudotime estimator, but this is left for future development.

Other interesting future research topics include applying BINGO to solve different biological and biomedical real-data problems. Some theory on GPDMs is presented in Supplementary Notes 1–3. From a theoretical perspective, it would be desirable to relax smoothness requirements and to consider process noise with memory and/or dependence on the system's state, which is also more realistic from the application point of view[42].

## Methods

**Theory on continuous-time GPDMs.** The existence and uniqueness of the solutions of the stochastic differential Eq. (2) are proven in Supplementary Note 1. The method's practical implementation is based on discretising Eq. (2) on a partition $\pi = \{0 = \tau_0 < \tau_1 < \ldots < \tau_M = T\}$ of the interval of interest $[0, T]$ using the (continuous) Euler scheme

$$\widehat{\mathbf{X}}_t = \widehat{\mathbf{X}}_{\tau_{k-1}} + (t - \tau_{k-1})\mathbf{f}(\mathbf{u}_{\tau_{k-1}}, \widehat{\mathbf{X}}_{\tau_{k-1}}) + \mathbf{w}_t - \mathbf{w}_{\tau_{k-1}}, \tag{8}$$

for $t \in [\tau_{k-1}, \tau_k]$ where $k = 1, \ldots, M$. In Supplementary Note 2, it is shown that $\widehat{\mathbf{X}}$ converges to the solution of Eq. (2), as the time discretisation is refined.

**Probability distribution of the trajectory.** The probability distribution $p(\mathbf{X}|\theta)$ is derived in this section for the discrete trajectory $\mathbf{X} = [\mathbf{X}_{\tau_0}, \ldots, \mathbf{X}_{\tau_M}]$, where $\theta$ denotes collectively all the hyperparameters. The discrete trajectory is sampled from the continuous version $\widehat{\mathbf{X}}_t$ given in Eq. (8), such that $\mathbf{X}_{\tau_k} = \widehat{\mathbf{X}}_{\tau_k}$. That is, it satisfies

$$\mathbf{X}_{\tau_k} = \mathbf{X}_{\tau_{k-1}} + \delta\tau_k \mathbf{f}(\mathbf{u}_{\tau_{k-1}}, \mathbf{X}_{\tau_{k-1}}) + \mathbf{w}_{\tau_k} - \mathbf{w}_{\tau_{k-1}}, \tag{9}$$

where $\delta\tau_k := \tau_k - \tau_{k-1}$, and $k = 1, \ldots, M$. It holds that

$$p(\mathbf{X}|\theta) = \int p(\mathbf{X}|\mathbf{f}, \theta)p(\mathbf{f}|\theta)d\mathbf{f}. \tag{10}$$

For given $\mathbf{f}$, the trajectory $\mathbf{X}$ is a Markov process, and therefore its distribution satisfies

$$p(\mathbf{X}|\mathbf{f}, \theta) = p(\mathbf{X}_{\tau_0}|\theta) \prod_{k=1}^{M} p(\mathbf{X}_{\tau_k}|\mathbf{X}_{\tau_{k-1}}, \mathbf{f}, \theta), \tag{11}$$

where

$$p(\mathbf{X}_{\tau_k}|\mathbf{X}_{\tau_{k-1}}, \mathbf{f}, \theta) = \frac{1}{(2\pi\delta\tau_k)^{n/2}|\mathbf{Q}|^{1/2}}$$
$$\cdot \exp\left(-\frac{1}{2\delta\tau_k}\left|\mathbf{X}_{\tau_k} - \mathbf{X}_{\tau_{k-1}} - \delta\tau_k\mathbf{f}(\mathbf{X}_{\tau_{k-1}})\right|^2_{\mathbf{Q}^{-1}}\right). \tag{12}$$

The integral in Eq. (10) can be computed analytically (more details can be found in Supplementary Note 3). After some more manipulation, the probability distribution for the discretised trajectory is obtained:

$$p(\mathbf{X}|\theta) = \frac{p(\mathbf{X}_{\tau_0}|\theta)}{(2\pi)^{Mn/2}} \prod_{i=1}^{n} \frac{1}{|\Delta\tau\mathbf{K}_i(\underline{\mathbf{X}})\Delta\tau + q_i\Delta\tau|^{1/2}}$$
$$\cdot \exp\left(-\frac{1}{2}(\overline{\mathbf{X}}_i - \underline{\mathbf{X}}_i)^\top(\Delta\tau\mathbf{K}_i(\underline{\mathbf{X}})\Delta\tau + q_i\Delta\tau)^{-1}(\overline{\mathbf{X}}_i - \underline{\mathbf{X}}_i)\right), \tag{13}$$

where $\Delta\tau$ is a diagonal matrix whose element $(k, k)$ is $\delta\tau_k$, $\overline{\mathbf{X}} := [\mathbf{X}_{\tau_1}, \ldots, \mathbf{X}_{\tau_M}]^\top$, and $\underline{\mathbf{X}} := [\mathbf{X}_{\tau_0}, \ldots, \mathbf{X}_{\tau_{M-1}}]^\top$. Same notation is also used for the different dimensions of the trajectory.

Note that $p(\mathbf{X}|\theta)$ corresponds to the finite-dimensional distribution of the continuous Euler-discretised trajectory $\widehat{\mathbf{X}}$ satisfying Eq. (8), evaluated at discretisation points. In Supplementary Note 2, it is shown that $\widehat{\mathbf{X}}$ converges strongly to the solution $\mathbf{x}$ of Eq. (2), and therefore the finite dimensional distributions converge as well. This means that $p(\mathbf{X}|\theta)$ is a proper finite dimensional approximation of the distribution of $\mathbf{x}$.

**Network inference method.** Consider then the original problem, that is, estimating the hyperparameters from given time series data. Denote $\mathbf{Y} = [\mathbf{y}_0, \mathbf{y}_1, \ldots, \mathbf{y}_m]$ where $\mathbf{y}_j$ is assumed to be a noisy sample from the continuous trajectory $\mathbf{x}$, that is, $\mathbf{y}_j = \mathbf{x}(t_j) + \mathbf{v}_j$, and $\mathbf{v}_j \sim \mathcal{N}(0, \mathbf{R})$ where $\mathbf{R} = \mathrm{diag}(r_1, \ldots, r_n)$. The intention is to

draw samples from the parameter posterior distribution using an MCMC scheme. Therefore, the posterior distribution is needed only up to constant multiplication. Denoting the hyperparameters collectively by $\theta$, the hyperparameter posterior distribution satisfies $p(\theta|\mathbf{Y}) \propto p(\mathbf{Y}, \theta) = \int p(\mathbf{Y}, \mathbf{x}, \theta)d\mathbf{x} = \int p(\mathbf{Y}|\mathbf{x}, \theta)p(\mathbf{x}|\theta)p(\theta)d\mathbf{x}$. Here $p(\mathbf{Y}|\mathbf{x}, \theta)$ is the Gaussian measurement error distribution, $p(\mathbf{x}|\theta)$ will be approximated by $p(\mathbf{X}|\theta)$ given in Eq. (13) for the discretised trajectory $\mathbf{X}$, and $p(\theta)$ is a prior for the hyperparameters. This prior consists of independent priors for each parameter. The integration with respect to the trajectory $\mathbf{x}$ is done by MCMC sampling. The function $f_i$ has mean $m_i(\mathbf{x}) = b_i - a_i x_i$ where $a_i$ and $b_i$ are regarded as hyperparameters corresponding to basal transcription ($b_i \geq 0$) and mRNA degradation ($a_i \geq 0$), and the squared exponential covariance

$$k_i(\mathbf{x}, \mathbf{z}) = \gamma_i \exp\left(-\sum_{j=1}^{n} \beta_{i,j}(x_j - z_j)^2\right). \tag{14}$$

The covariance hyperparameters satisfy $\gamma_i > 0$ and $\beta_{i,j} \geq 0$. The network inference is based on estimating the parameters $\beta_{i,j}$. If $\beta_{i,j} > 0$, the function $f_i$ can depend on $x_j$. In the context of GRN inference, it indicates that gene $j$ is a regulator of gene $i$.

For the MCMC sampling of the hyperparameters $\beta_{i,j}$, indicator variables are used[48]. That is, each of them is represented as a product $\beta_{i,j} = S_{i,j}H_{i,j}$, where $S_{i,j} \in \{0, 1\}$ and $H_{i,j} \geq 0$. The state of the sampler consists of the indicator variable $\mathbf{S}$, the hyperparameters $H_{i,j}$, $\gamma_i$, $r_i$, $q_i$, $a_i$, $b_i$ ($i, j = 1, \ldots, n$) and the discrete trajectory $\mathbf{X}$. The trajectory is sampled using a Crank–Nicolson sampler[49,50] (see Supplementary Note 7). One row of the indicator matrix $\mathbf{S}$ is sampled by randomly choosing one element, and changing it from zero to one, or from one to zero. The other variables are sampled using a Metropolis–Hastings within Gibbs sampler with random walk proposals. A pseudo-input scheme[51,52] is used to speed up the Gaussian process framework (see Supplementary Note 6).

The full algorithm is presented in Supplementary Note 4 (and Supplementary Table 1). In the drawn samples, the network structure information is in the indicator variable samples $\mathbf{S}^{(l)}$, where $l$ refers to the number of the sample. The output of the algorithm is the average of these samples, which converges as more and more samples are collected:

$$\frac{1}{N_{\mathrm{sample}}} \sum_{l=1}^{N_{\mathrm{sample}}} \mathbf{S}^{(l)} \to \mathbb{E}(\mathbf{S}|\mathbf{Y}). \tag{15}$$

Since $\mathbf{S}$ is zero-one valued, the element $(i, j)$ of the matrix $\mathbb{E}(\mathbf{S}|\mathbf{Y})$ is the posterior probability that $\beta_{i,j}$ is not zero, given the data $\mathbf{Y}$.

For $q_i$, $r_i$, and $\gamma_i$ we use noninformative inverse Gamma priors. For $H_{i,j}$, $a_i$, and $b_i$ we use Laplace priors (exponential), and for $\mathbf{S}$ we use $p(\mathbf{S}) \propto \eta^{|\mathbf{S}|_0}$ where $|\mathbf{S}|_0$ gives the number of ones in $\mathbf{S}$. This prior means that the existence of a link is a priori independent of the existence of other links, and each link exists with probability $p = \frac{\eta}{\eta+1}$. The parameter $\eta \geq 0$ can be set to obtain a desired sparsity level for the samples. It is the only user-defined parameter in the method—save for discretisation steps and sampler step lengths. Prior distribution specifications are discussed in more detail in Supplementary Note 8. Other simulation details and computation times for the benchmark cases are presented in Supplementary Tables 2 and 3.

Several time series experiments, including knockout/knockdown experiments, can be easily incorporated into the time series by concatenating the time series. Steady state measurements are included by adding them into the dataset as points where $\mathbf{f}(\mathbf{x}_{ss}) = 0$. These procedures are described in Supplementary Note 5. It should be noted, however, that the concatenated time series should have similar properties. If this is not the case, for example if the different time series are obtained by different experimental techniques, or the pre-processing is different, then it is advised to use the method separately for each dataset. External inputs, such as the perturbations (with unknown targets) in the DREAM4 data can be included as additional state variables, which are not sampled. This feature is included in the method's numerical implementation. If certain genes are interesting as potential regulators, but not as target genes, they can be included as inputs to reduce the computational burden of the method.

**Data processing.** The method can handle data from any experimental technique producing bulk time series data. It is assumed in the method that the time series data have been pre-processed appropriately, depending on the used experimental technique[53–55]. In particular, measurements between time points should be comparable, meaning that normalisation with respect to housekeeping genes should be carried out. By default, the BINGO code normalises the data by scaling the dynamical range of each gene ($\max_j [\mathbf{y}_j]_i - \min_j [\mathbf{y}_j]_i$) to one. In order to avoid noise amplification, it is important to exclude all genes from the data that are not properly expressed. Also genes that seem to be mostly noise, should be removed.

Pre-processing steps often include background removal and nonlinear transformations (e.g., taking logarithms). Moreover, microarray data are based on luminescence measurements which has a nonlinear dependency on the mRNA concentration. Constant shifts of the data have very little effect on BINGO, since the probability distribution in Eq. (13) only depends on differences of gene expression values between different time points. The only small effect of shifting data comes from the GP mean $m_i(\mathbf{x}) = b_i - a_i x_i$ where a constant shift has an effect on the priors for $a_i$ and $b_i$. Nonlinear transformation has a more intricate effect. Say

$z_i = g(x_i)$ with some smooth, invertible function $g$. If $dx_i = f_i(\mathbf{x})dt + dw_i$, then

$$dz_i = g'(g^{-1}(z_i))f_i(g^{-1}(z_1), \dots, g^{-1}(z_n))dt + g'(g^{-1}(z_i))dw_i. \quad (16)$$

The GRN topology is not affected by the nonlinear transformation, but the functional form, and the process noise characteristics are affected. The GP framework is flexible, and it should be able to fit to both $f_i(\mathbf{x})$ and $g'(g^{-1}(z_i))f_i(g^{-1}(z))$.

As a general rule, BINGO should work best when the data correspond as closely as possible to the actual expression levels (that is, mRNA concentrations). However, if the data are very spiky, that is, concentrations peak very high on few measurement times, then a log-transformation might be beneficial, since BINGO assumes smooth dynamics functions.

**Reporting summary**. Further information on research design is available in the Nature Research Reporting Summary linked to this article.

## Data availability

The four different datasets used in the article are available as follows:
- DREAM4 in silico network inference challenge data[2,23,24,39] are available at https://www.synapse.org/#!Synapse:syn3049712/wiki/74630.
- Simulated data from *Arabidopsis thaliana* circadian clock are available as example data along with the method's code.
- In vivo dataset IRMA[26] are available as supplemental data for the article, https://doi.org/10.1016/j.cell.2009.01.055.
- Data from *Arabidopsis thaliana* circadian clock[44] are available at: http://www.ncbi.nlm.nih.gov/geo/query/acc.cgi?acc=GSE19271.

## Code availability

The method's MATLAB implementation is available at https://github.com/AtteAalto/BINGO.

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

## Acknowledgements

A.A. was supported in part by ERASysApp and Fonds National de la Recherche (FNR) Luxembourg reference INTER/SYSAPP/14/02, by University of Luxembourg Internal Research Project reference OptBioSys, and by FNR Luxembourg CORE Junior reference C19/BM/13684479. J.G. was partly supported by the 111 Project on Computational Intelligence and Intelligent Control under Grant B18024.

## Author contributions

A.A. has developed the method and done the numerical implementation. The theory of the GPDM was developed by L.V. and he wrote Supplementary Notes 1 and 2. Major part of the main manuscript was written by A.A., with participation of P.I., L.M., and J.G. L.M. did part of the comparison studies, processed the real *A. thaliana* data, and wrote code for some data management and simulation of the Millar10 model.

## Competing interests

The authors declare no competing interests.
