## [Peer Review File · Nature Communications]

Reviewers' comments:

Reviewer #1 (Remarks to the Author):

The paper presents an interesting approach to inferring gene regulatory networks from transcriptomic data time series by modelling an underlying stochastic process from which observations are generated. The paper is well written and the method appears to outperform many existing approaches. The method is implemented as an open source software package. The contributions made by the authors could be clearer, and some of the evaluation of the method could be improved.

Major comments:

- The novelty in the work could be stated more clearly and concisely - how does this work expand on the GPDM? The authors should also explain briefly how this approach differs from existing Gaussian Process methods such as CSI.
- In networks like gene regulatory networks, that are sparse, there is typically a large class imbalance between the positive and negative classes, as the majority of possible edges do not exist. This can lead to measures such as AUROC and AUPR performing poorly. It is also often the case that final inferred networks are generated at a single cut-off (here of the posterior probability of an edge existing), so that performance over the whole ROC curve or PR curve is less relevant, and numbers of TP and FP edges identified at a realistic cut-off are more informative. It would be interesting to also consider measures such as the Matthews Correlation Coefficient, which is more robust under a large class imbalance.
- The DREAM4 challenge used in evaluation simulates microarray data, whereas modern data such as the single cell data the authors mention is typically collected using RNA-Seq based methods. These produce data that have very different properties to microarray data, and there should be some validation on such data or a discussion of which forms of transcriptomic data the method can be applied to, and what pre-processing steps are needed.
- Some of the statements made in the introduction are either inaccurate or not clear. The authors discuss model based inference of networks, but focus on methods that simulate trajectories and compare them to observed data, when there are a number of papers that apply statistical models and infer networks using MCMC or related techniques, without resorting to simulations. The authors also appear to state that it is not possible to simulate trajectories from a Gaussian process, and that nonparametric models mostly work by estimating derivatives. However information theoretic approaches, whilst nonparametric, are able to infer GRNs without using derivatives[1]. The introduction also implies that simulation and nonparametric methods based on derivatives are the main approaches used, when graphical models are one of the major classes of GRN inference methods in the literature. The authors also do not appear to reference any works applying derivatives to learn GRNs.

Minor comments:

- There is no application of the method to a large scale experimental data set. The IRMA data set consists only of five genes. Although there may be no gold standard to benchmark against, it could improve the impact of the paper to demonstrate the results that can be produced from application to a larger number of genes from e.g. a recent RNA-Seq time series data set.

[1]Thalia E. Chan, Michael P.H. Stumpf, Ann C. Babbie, Gene Regulatory Network Inference from Single-Cell Data Using Multivariate Information Measures, Cell Systems, Volume 5, Issue 3, 2017.

Reviewer #2 (Remarks to the Author):

This paper proposed a method named BINGO based on GPDM to infer the gene regulatory network from low-sampled time-series gene expression dataset. It showed better results on DREAM4 and IRMA datasets than several other algorithms. However, the manuscript was not well written, and there are many issues unclear.

1. The major contribution of this manuscript is the trajectory sampling, which artificially generates new time-points (interpolation) between the measured times so as to alleviate the problem of the low sampling frequency and overall small number of samples. By iterating the trajectory sampling and the topology sampling, BINGO obtains the gene regulatory network, which is consistent with both the measured data and newly generated data. It is interesting, but it is unclear if or not the artificially generated data is additional information (from other sources) and simply derived from the measured data. If it is not additional data, BINGO is self-consistent and still has the problem of the overfitting for the small number of samples.
2. It is very difficult to understand the theoretical descriptions in the manuscript. In particular, for Algorithm 2.1 in SI, there are so many symbols and variables or criteria undefined or unexplained.
3. Fig.2 shows the performance of BINGO under different sample rates, but it is unclear which data is high-rate sampling and which is low-rate sampling.
4. Fig.2 and table 2 show the data with high noise data has better AUROC/AUPR than that with low noise data. Why? How about the results without the noise?
5. Transfer entropy and many other information-based algorithms are simple and fast. Could the author compare BINGO with them?
6. How fast is BINGO for data with different size? And other methods? Is there an example can show the advantage of BINGO on low-sampled data, while performs almost the same with other algorithms for high-sampled data?

Reviewer #1 (Remarks to the Author):

The paper presents an interesting approach to inferring gene regulatory networks from transcriptomic data time series by modelling an underlying stochastic process from which observations are generated. The paper is well written and the method appears to outperform many existing approaches. The method is implemented as an open source software package. The contributions made by the authors could be clearer, and some of the evaluation of the method could be improved.

Major comments:

- The novelty in the work could be stated more clearly and concisely - how does this work expand on the GPDM? The authors should also explain briefly how this approach differs from existing Gaussian Process methods such as CSI.

Response: We have rewritten large parts of the introduction to better highlight the novelty of the work. In particular, the paragraph in the introduction where BINGO is first mentioned (end of page 2) now explicitly and concisely states the novelty of BINGO: to introduce statistical trajectory sampling that enables continuous-time nonparametric ODE approaches. Regarding Gaussian process dynamical models (GPDMs), BINGO uses GPDM as a model class that captures gene expression dynamics. While discrete-time GPDMs have been introduced in the literature, there are no continuous-time versions. Molecular networks are dynamic and evolve in continuous-time and therefore models need to capture this property, especially with low sampling rates, typically found in biology. Hence, one contribution of this work is the development of continuous-time GPDMs, defined as stochastic differential equations. Some properties of continuous-time GPDMs are established in supplementary information. This is now mentioned in the beginning of Section 1.2.

A paragraph has been added to the introduction to clarify key differences between BINGO and existing Gaussian process network inference methods. Basically, existing GP approaches, such as CSI, treat network inference problems in two steps: derivative estimation, and network inference with similar tools as discrete-time methods (e.g. LASSO). The major problem here is that derivative estimation directly from data is impaired by low sampling rates, and therefore should be avoided. BINGO, on the other hand, is based on deriving probability distributions for continuous trajectories and sampling from these distributions, hence avoiding derivative estimation.

- In networks like gene regulatory networks, that are sparse, there is typically a large class imbalance between the positive and negative classes, as the majority of possible edges do not exist. This can lead to measures such as AUROC and AUPR performing poorly. It is also often the case that final inferred networks are generated at a single cut-off (here of the posterior probability of an edge existing), so that performance over the whole ROC curve or PR curve is less relevant, and numbers of TP and FP edges identified at a realistic cut-off are more informative. It would be interesting to also consider measures such as the Matthews Correlation Coefficient, which is more robust under a large class imbalance.

Figure 1: Left: MCC computed for different methods and different thresholds for network 1 in the DREAM4 data with size 10. Center: Comparison between AUPR score and MCC for the four methods in all 5 networks of the DREAM4 data with size 10 together with the least squares regression line in red. Right: Correlation between the MCC and AUPR scores as function of number of links used to compute MCC.

Response: We agree with the reviewer’s comments on class imbalance and threshold selection problem. However, AUROC and AUPR values are objective performance measures for network inference methods, and they can be regarded as standard performance metrics in network inference. For example, these scores have been used in the DREAM4 challenge, in the review article by Penfold and Wild (2011), and in Casadiego et al. (2017), to name a few. It is true that the AUROC gives too much weight on low confidence predictions. This, however, is not the case with the AUPR score that gives a strong weight on those predictions that the methods assign high confidence. Moreover, some results taken from the literature, for comparison with BINGO, only have AUROC/AUPR scores available. Hence, for those it is not possible to compute other performance scores without first obtaining their code. A discussion on these performance metrics has been added to the beginning of Section 2.

To verify BINGO’s performance using MCC, we computed MCCs for four methods in the comparison for the DREAM4 networks of size 10 (iCheMA was excluded since we only stored the resulting AUROC/AUPR values for that method). MCCs were computed by considering networks with a fixed number of links with highest confidence. These results are illustrated in Figure 1 above. The left panel shows MCCs computed using different number of links for the first network. BINGO’s performance is particularly good for links that the method assigns high confidence. This is also reflected by AUPR scores for this particular network: BINGO, dynGENIE3, ARNI, and GRNTE are 0.829, 0.551, 0.682, and 0.481, respectively. The middle panel shows AUPRs versus MCCs computed for networks with the 15 highest confidence links. The correlation between AUPRs and MCCs is 0.847. The right panel shows the correlation between AUPRs and MCCs as a function of the number of highest confidence links used to compute MCC. For small number of links, the correlation is between 0.84 and 0.9, meaning that AUPRs and MCCs are highly correlated, i.e. contain similar information. As the number of links increases and become less confident (more arbitrary), the correlation only decreases slightly.

In summary, and for the reasons stated above, in this particular application MCC did not add much information and, hence, we decided to continue using AUROC and AUPR.

Typically, the choice of a cut-off threshold depends on the end user. An alternative to choosing a particular threshold is to initially set conservative thresholds to obtain high confidence links. Then, further links obtained with lower thresholds are less credible, but may still give valuable hints at potential gene interactions to be studied further. Indeed, this is exactly what BINGO gives the user: probabilities that particular links exist. The end user can either set a threshold or consider each link with the confidence given by its associated probability.

- The DREAM4 challenge used in evaluation simulates microarray data, whereas modern data such as the single cell data the authors mention is typically collected using RNA-Seq based methods. These produce data that have very different properties to microarray data, and there should be some validation on such data or a discussion of which forms of transcriptomic data the method can be applied to, and what pre-processing steps are needed.

Response: The vast majority of network inference methods, including BINGO, take pre-processed data as inputs. Hence, they can be applied to any type of time-series data, including RNA-Seq, microarray, protein levels, etc.

The step of pre-processing data is indeed fundamental before performing any modelling. There are large numbers of pre-processing algorithms in the literature for just about any type of experimental data, including microarray (Gyorffy et al., 2009) and RNA-seq (Everaert et al., 2017). Indeed, RNA-seq data are reported to have larger dynamical range and higher accuracy than microarray data (Zhao et al., 2014), which should be beneficial for network inference.

A discussion on data pre-processing has been added to the revised manuscript (Section A.4), where we refer to the aforementioned articles on pre-processing for different experimental techniques. In addition, we discuss effects of shifting data and general nonlinear transformations (such as log-transformation, nonlinear dependency between luminescence and mRNA concentration in microarray experiments). Also, genes that are hardly expressed were removed from the data to avoid confusing dynamics with noise (this is now mentioned in the readme-file of the code in Github, and in the Appendix of the manuscript).

For the specific data from DREAM4, they included measurement noise (a combination of normally distributed and log-normally distributed noise) to mimic microarray measurements. BINGO assumes noise to be normally distributed, so no particular technique-specific assumptions have been made. Measurement error variance is estimated automatically, and therefore BINGO should adapt to different types of data without modifications.

- Some of the statements made in the introduction are either inaccurate or not clear. The authors discuss model based inference of networks, but focus on methods that simulate trajectories and compare them to observed data, when there are a number of papers that apply statistical models and infer networks using MCMC or related techniques, without resorting to simulations.

Response: We agree with the reviewer that the discussion on GRN inference approaches in the introduction was confusing. Indeed, the introduction of the previous version mainly focused on identifying ODE models for gene expression (and, in particular, ODE models with nonparametric function estimators). The reviewer is also correct that some statements were not written accurately enough. The corresponding two paragraphs in the introduction have now been rewritten. The introduction now begins by mentioning other general approaches (information theoretic methods, Bayesian networks), before focusing on ODE-based methods.

Following a suggestion from the other reviewer, we have included a transfer entropy based method into the comparison.

The authors also appear to state that it is not possible to simulate trajectories from a Gaussian process,

Response: We have removed this claim. In general, however, it is usually not feasible to simulate trajectories from nonparametric ODE models, such as Gaussian process dynamical models (GPDM). A direct trajectory simulation from the GPDM would require first sampling a Gaussian process realisation f , and then simulating a trajectory satisfying $dx/dt = f(x) + w$.

The authors also appear to state that [...] nonparametric models mostly work by estimating derivatives. However information theoretic approaches, whilst nonparametric, are able to infer GRNs without using derivatives[1].

Response: As discussed above, this claim was concerned with nonparametric ODE-models (Gaussian process, random forest), and the claim has been clarified in the revised introduction. Other GRN inference approaches are now discussed in the introduction, including information theoretic approaches, such as [1].

The authors also do not appear to reference any works applying derivatives to learn GRNs.

Response: The division to methods based on input-output regression (from estimated derivatives) and to methods based on simulating trajectories is now clarified, and all references are placed accordingly, indicating methods that estimate gene expression derivatives. Different derivative estimation strategies are also specified (difference approximation, spline fitting, Gaussian process fitting, filtering) in the paragraph halfway through p. 2.

Minor comments:

- There is no application of the method to a large scale experimental data set. The IRMA data set consists only of five genes. Although there may be no gold standard to benchmark against, it could improve the impact of the paper to demonstrate the results that can be produced from application to a larger number of genes from e.g. a recent RNA-Seq time series data set.

Response: We have now added another real data example (Section 2.4), where BINGO is applied on a dataset of a recent study on the effect of nicotinamide (NAM)

on the circadian clock of *Arabidopsis thaliana* by Mombaerts *et al.* (2019). Two experiments have been carried out with this data. In the first experiment, BINGO was applied on a set of 17 known circadian clock genes. In this experiment, we are able to identify the same genes as potential NAM targets as reported by Mombaerts *et al.* In the second example, BINGO was applied on 994 genes that were identified periodic in both NAM-treated and untreated plants. This experiment is done mainly to demonstrate BINGO's scalability to higher dimensions, although the results have the potential to reveal novel gene interactions.

[1] Thalia E. Chan, Michael P.H. Stumpf, Ann C. Babbie, Gene Regulatory Network Inference from Single-Cell Data Using Multivariate Information Measures, Cell Systems, Volume 5, Issue 3, 2017.

Reviewer #2 (Remarks to the Author):

This paper proposed a method named BINGO based on GPDM to infer the gene regulatory network from low-sampled time-series gene expression dataset. It showed better results on DREAM4 and IRMA datasets than several other algorithms. However, the manuscript was not well written, and there are many issues unclear.

1. The major contribution of this manuscript is the trajectory sampling, which artificially generates new time-points (interpolation) between the measured times so as to alleviate the problem of the low sampling frequency and overall small number of samples. By iterating the trajectory sampling and the topology sampling, BINGO obtains the gene regulatory network, which is consistent with both the measured data and newly generated data. It is interesting, but it is unclear if or not the artificially generated data is additional information (from other sources) and simply derived from the measured data. If it is not additional data, BINGO is self-consistent and still has the problem of the overfitting for the small number of samples.

Response: BINGO is self-contained, that is, no external information is used in sampling the trajectories between measurement points. The sampling scheme draws samples from the probability distribution derived for the continuous trajectories of the Gaussian process dynamical model (conditioned on the observations). The model contains the GRN topology encoded in the hyperparameters of the Gaussian process. The topology is sampled as well. Overfitting is avoided, since BINGO does not fit a complicated model with a high number of free parameters to the data, and instead focuses only on the (sparse) network topology, while all model (hyper)parameters are integrated out by MCMC sampling.

We have clarified this point with further discussions on trajectory sampling after equation (1.3), and a discussion on overfitting in the introduction (middle of p. 3).

2. It is very difficult to understand the theoretical descriptions in the manuscript. In

particular, for Algorithm 2.1 in SI, there are so many symbols and variables or criteria undefined or unexplained.

Response: Indeed, the supplementary information was not really written as a standalone document, which understandably made it difficult to follow. Section 2 in the SI now begins with an explanation of the algorithm, and all symbols are properly introduced. Section 1.3 in the SI has also been improved in this respect. Algorithm 2.1 now refers to Table 1 containing a list of parameters and variables with brief descriptions.

3. Fig.2 shows the performance of BINGO under different sample rates, but it is unclear which data is high-rate sampling and which is low-rate sampling.

Response: Different sampling rates are now indicated in the figure, as well as the predictions from full and averaged data for the IRMA example.

4. Fig.2 and table 2 show the data with high noise data has better AUROC/AUPR than that with low noise data. Why ? How about the results without the noise ?

Response: For most methods, the results improve with higher process noise (not measurement noise), at least in the cases of 1h and 2h sampling rates. Process noise excites the system, providing new dynamics that contain information beneficial for modelling. The *Arabidopsis thaliana* data is collected following the transition period from light/dark cycles to constant light condition. Hence, the resulting system dynamics are a consequence of both nonzero initial states and process noise, and both of these contribute to identification. This is now mentioned in the article where these results are discussed.

To test the performance without noise, we ran one more experiment (using BINGO and dynGENIE3) on a noise-free setup with 2h sampling rate. In this experiment, BINGO achieved AUROC 0.877 and AUPR 0.869, and dynGENIE3 achieved AUROC 0.635 and AUPR 0.403. Interestingly, BINGO's results are comparable to the those with high process noise. These results are not included in the manuscript since noise-free setups are unrealistic.

5. Transfer entropy and many other information-based algorithms are simple and fast. Could the author compare BINGO with them?

Response: A new transfer entropy based network inference method (GRNTE) has been added to the comparisons. It has reasonable performance with the DREAM4 data but not very satisfactory with the *A. thaliana* circadian clock nor with the IRMA data. Please see Figure 2 and Tables 1–3 in the main text for the complete results.

6. How fast is BINGO for data with different size? And other methods? Is there an example can show the advantage of BINGO on low-sampled data, while performs almost the same with other algorithms for high-sampled data?

Response: Computation times of BINGO for all example cases are given in Table 2 of supplementary information, and for all methods in one example in Table 3 of supplementary information. The computational costs are discussed in the Discussion-section of the article. While it is true that BINGO is slower than most other methods in this comparison (with the exception of iCheMA), it is still computationally feasible to dimension up to 2000. The new, 994-dimensional nicotinamide example took 19 hours to complete on a desktop workstation utilising 20 parallel processors. This could still be improved by parallelising further using high performance computing clusters. It should be noted that GRN inference methods are not intended for real-time use, and so computational time is not of primary concern, as long as the methods are feasible for large enough problems.

As for the second question, in the *Arabidopsis thaliana* example, BINGO's performance with 4h sampling rate is better than any other method's performance with 1h sampling rate. Moreover, most methods' performances deteriorate clearly when going from 2h to 4h sampling rate, whereas BINGO's (and dynGENIE3's) performance does not suffer badly.

References:

J. Casadiego, M. Nitzan, S. Hallerberg, and M. Timme: "Model-free inference of direct network interactions from nonlinear collective dynamics", *Nature Communications* 8:2192, (2017).

C. Everaert, M. Luybaert, J.L.V. Maag, Q.X. Cheng, M.E. Dinger, J. Hellemans, and P. Mestdagh: "Benchmarking of RNA-sequencing analysis workflows using whole-transcriptome RT-qPCR expression data", *Scientific Reports* 7:1559, 2017.

B. Gyorffy, B. Molnar, H. Lage, Z. Szallasi, and A.C. Eklund: "Evaluation of microarray preprocessing algorithms based on concordance with RT-PCR in clinical samples", *PLoS ONE* 4(5):e5645, 2009.

L. Mombaerts, A. Carignano, F.C. Robertson, T.J. Hearn, J. Junyang, D. Hayden, Z. Rutterford, C.T. Hotta, K.E. Hubbard, M.R.C. Maria, Y. Yuan, M.A. Hannah, J. Goncalves, and A.A.R. Webb: "Dynamical differential expression (DyDE) reveals the periodic control mechanisms of the *Arabidopsis* circadian oscillator", *PLoS Computational Biology* 15(1):e1006674, (2019).

C.A. Penfold and D.L. Wild: "How to infer gene networks from expression profiles", *Interface Focus* 1(6):857-870, (2011).

S. Zhao, W.-P. Fung-Leung, A. Bittner, and X. Liu: "Comparison of RNA-Seq and microarray in transcriptome profiling of activated T cells", *PLoS ONE* 9(1): e78644, (2014).

REVIEWERS' COMMENTS:

Reviewer #1 (Remarks to the Author):

This updated version of the paper has a much improved introduction that clearly places the method in the context of other work in the field.

The application to a larger data set, addition of information on preprocessing and thorough exploration of the value of adding the MCC benchmark addresses my original comments about the paper.

In the NAM study, figure 4 could also include the full distribution of posterior edge probabilities for the larger gene set, as well as only the NAM targets shown in (c). The cut-off of 0.15 to select edges seems low, especially compared to that used in the 17 gene data set. It would be interesting to see if the larger gene set results in an overall lower edge posterior probability distribution, and not only in the NAM targets shown.

Reviewer #2 (Remarks to the Author):

The manuscript has significantly been improved. I still have a few comments. (1). This work uses MCMC rather than simulating the SDE, to give perturbation to the sampled trajectory. Bayesian inference is more essential than its so-called dynamical mechanism. (2). The low accepted rate of the MCMC is one problem. It is usually very low, and thus would take long-time computation for a large-scale problem. How to better utilize the posterior distribution $p(\theta|Y)$ is one important issue.

Responses to reviewers' comments

Reviewer #1 (Remarks to the Author):

This updated version of the paper has a much improved introduction that clearly places the method in the context of other work in the field.

The application to a larger data set, addition of information on preprocessing and thorough exploration of the value of adding the MCC benchmark addresses my original comments about the paper.

In the NAM study, figure 4 could also include the full distribution of posterior edge probabilities for the larger gene set, as well as only the NAM targets shown in (c). The cut-off of 0.15 to select edges seems low, especially compared to that used in the 17 gene data set. It would be interesting to see if the larger gene set results in an overall lower edge posterior probability distribution, and not only in the NAM targets shown.

Response: The full distribution of posterior edge probabilities has been added (Figure 4d). It is true that in a larger set of genes, there are many genes with similar expression profiles over time and the posterior probabilities are somewhat diluted between genes with similar expression.

Reviewer #2 (Remarks to the Author):

The manuscript has significantly been improved. I still have a few comments. (1). This work uses MCMC rather than simulating the SDE, to give perturbation to the sampled trajectory. Bayesian inference is more essential than its so-called dynamical mechanism. (2). The low accepted rate of the MCMC is one problem. It is usually very low, and thus would take long-time computation for a large-scale problem. How to better utilize the posterior distribution $p(\theta|Y)$ is one important issue.

Response: It is true that from the perspective of Bayesian inference, it does not matter how the trajectories are generated. However, the trajectory sampling is a feasible way to obtain trajectories from a Gaussian process dynamical system. Sampling first the Gaussian process and then simulating trajectories does not seem feasible in our opinion. The acceptance probabilities are often low, in particular in the case of the discrete-valued network topology. However, parallelisation helps considerably, when the problem is large. Parallel chains are independent, and so they help to achieve a more complete picture of the full posterior distribution compared to what would be obtained by a single chain when the acceptance rate is low.